# *Eph* and *Ephrin* Variants in Malaysian Neural Tube Defect Families

**DOI:** 10.3390/genes13060952

**Published:** 2022-05-26

**Authors:** Siti Waheeda Mohd-Zin, Amelia Cheng Wei Tan, Wahib M. Atroosh, Meow-Keong Thong, Abu Bakar Azizi, Nicholas D. E. Greene, Noraishah Mydin Abdul-Aziz

**Affiliations:** 1Invertebrate and Vertebrate Neurobiology Laboratory, Department of Parasitology, Faculty of Medicine, Universiti Malaya, Kuala Lumpur 50603, Malaysia; mva190025@siswa.um.edu.my (S.W.M.-Z.); tcwamelia@gmail.com (A.C.W.T.); 2Department of Parasitology, Faculty of Medicine, Universiti Malaya, Kuala Lumpur 50603, Malaysia; wahib@um.edu.my; 3Department of Paediatrics, Faculty of Medicine, Universiti Malaya, Kuala Lumpur 50603, Malaysia; thongmk@um.edu.my; 4Neurosurgery Unit, Department of Surgery, Faculty of Medicine, Universiti Kebangsaan Malaysia, Cheras 56000, Malaysia; azizi@ppukm.ukm.edu.my; 5Developmental Biology and Cancer Department, Great Ormond Street Institute of Child Health, University College London, London WC1N 1EH, UK; n.greene@ucl.ac.uk

**Keywords:** *Eph*, *ephrins*, *EPHA2*, *EPHB6*, *EFNB1*, spina bifida, neural tube defects

## Abstract

Neural tube defects (NTDs) are common birth defects with a complex genetic etiology. Mouse genetic models have indicated a number of candidate genes, of which functional mutations in some have been found in human NTDs, usually in a heterozygous state. This study focuses on *Ephs*-*ephrins* as candidate genes of interest owing to growing evidence of the role of this gene family during neural tube closure in mouse models. *Eph*-*ephrin* genes were analyzed in 31 Malaysian individuals comprising seven individuals with sporadic spina bifida, 13 parents, one twin-sibling and 10 unrelated controls. Whole exome sequencing analysis and bioinformatic analysis were performed to identify variants in 22 known *Eph-ephrin* genes. We reported that three out of seven spina bifida probands and three out of thirteen family members carried a variant in either *EPHA2* (rs147977279), *EPHB6* (rs780569137) or *EFNB1* (rs772228172). Analysis of public databases shows that these variants are rare. In exome datasets of the probands and parents of the probands with *Eph*-*ephrin* variants, the genotypes of spina bifida-related genes were compared to investigate the probability of the gene–gene interaction in relation to environmental risk factors. We report the presence of *Eph*-*ephrin* gene variants that are prevalent in a small cohort of spina bifida patients in Malaysian families.

## 1. Introduction

Neural Tube Defects (NTDs) are a group of congenital malformations in which the formation of the brain and/or spinal cord are compromised as a result of failed closure of the embryonic neural tube, during the fourth week of post-fertilization development [1]. NTDs are among the most common congenital malformations and occur in 0.5 to 10 per 1000 pregnancies globally [2]. The clinical severity of NTDs varies [3,4], with open lesions affecting the brain (anencephaly) and craniorachischisis being the most severe phenotypes. Spina bifida describes lesions affecting the spinal region and affected individuals often exhibit motor and sensory neurological deficits below the level of the lesion [5,6]. In view of the life-changing health and economic consequences of NTDs, considerable effort should be invested in exploring the pathophysiological mechanisms that govern the etiology of NTDs in lieu of the ultimate goal of primary prevention [4].

Spina bifida is etiologically known for both genetic and environmental factors [2]. For instance, several nongenetic risk factors have been identified as having possibilities for prevention by way of maternal folic acid supplementation. On the other hand, genetic alterations have been found to cause NTDs in mice, with more than 250 genes having been shown to cause NTDs when mutated [7,8]. Planar cell polarity (PCP) genes are among the many genes implicated in the mechanism underlying neural tube closure [2,7]. So far, our knowledge of the causative genes in humans is less complete and around 59 genes have been reported to be potentially associated with spina bifida in humans [8,9,10] (Appendix A). Each gene has been implicated in only a small proportion of NTD patients or in specific populations [11,12,13,14], suggesting that there is considerable heterogeneity underlying the genetic basis of NTDs.

Ephs and ephrins are attractive candidates to be involved in NTDs in view of the involvement of Eph-ephrin interactions and signaling during adhesion and fusion processes [15,16,17,18,19,20,21,22]. Evidence of the possible involvement of Ephs and ephrins in neural tube development comes from neural tube defects models in mice [15,17,22], *Xenopus* [18,19] and zebrafish [21]. The first murine Eph and ephrin knockouts exhibited an open neural tube defect (ephrin-A5 null mice) [15], the second was a spina bifida occulta model (double heterozygous *EphA2^tmjrui/+^ EphA4^rb−2J/+^)* [22] and the most recent one (*Efnb1-*deficient embryo) was undefined as the phenotype was not assessed during the closure of primary neurulation [17]. In *Xenopus* embryo models, Eph and ephrin knockdown *(EphA7-*morpholino oligonucleotide (MO) and *ephrinB2-*MO) disrupt cranial neural tube closure [18,19]. Although neurulation does not occur by closure of neural folds, mosaic Eph and ephrin morpholino (loss-of-function) treatment of zebrafish embryos suggest that the Eph and ephrin are specifically and individually required to facilitate integration of progenitor cells during the cross-midline cell division that occurs at the neural keel [21]. Although teleost neural keel formation and mammalian neural tube closure are developmentally distinct events, defects in either process result in severe neural tube defects [1,23,24].

Ephs are the largest group of receptor tyrosine kinases (RTKs) and are known to not only bind to their ligand ephrins but Eph-ephrin complexes are also known to interact or couple with co-receptors such as the TrkB neurotrophin receptor [25,26,27], p75 neurotrophin receptor [28], and Ret tyrosine kinase receptor [29,30]. Eph-ephrin complexes can also activate or inhibit signaling effectors such as protein tyrosine phosphatase (PTPase) SHP2 protein [31,32,33], Ras/Rho family GTPases [34], ADP-ribosylation factor 6 (Arf6) [17], and focal adhesion kinase (FAK) [18]. Ephs are integral membrane receptors, whereas ephrinA ligands are linked to the plasma membrane via a cell surface glycosyl phosphatidylinositol (GPI)-anchor, and ephrinBs are transmembrane ligands. EphrinA5 is implicated in interactions with different splice forms of EphA7 to mediate cellular adhesion or repulsion during neural fold fusion in mouse embryos [15]. EphrinB1 is associated with the maintenance of the structural integrity of apical cell and extracellular matrix (ECM) adhesions in mouse neuroepithelial development [17]. Moreover, a functional role for Eph–ephrin interactions during neural tube closure was suggested by the finding that closure was delayed in mouse embryos cultured with EphA1 and EphA3 fusion proteins used to specifically disrupt ephrinA–EphA receptor interactions [16]. In mouse embryos, *EphA2* is shown to be expressed in a lamellipodium-like protrusion structure which extends towards the opposite neural fold [16].

In mouse knockout embryos with perturbation of both alleles of one gene and a single allele of the second gene simultaneously in *EphA2* and *EphA4* crosses (*Epha2^tm1Jrui/+^Epha4^rb−2J/rb−2J^* and *Epha2^tm1Jrui/tm1Jrui^Epha4^rb−2J/+^*), a large number of rounded cells were seen in the open cranial and open spinal neuropores [22]. Furthermore, the double heterozygous embryos carrying the loss of function alleles of *EphA2* and *EphA4* (*Epha2^tm1Jrui/+^Epha4^rb−2J/+^*) exhibit spina bifida occulta and exencephaly at a penetrance of more than 50%. These findings suggest a dual compensatory role of *EphA2* and *EphA4* during murine spinal neural tube closure [22]. To date there has been no systematic study to implicate *Ephs* and *ephrins* in a human spina bifida cohort although a multitude of Ephs and ephrins have been implicated in human cancers [35,36,37,38,39].

It is important to assess the genetic basis of NTDs in diverse populations. Progress has been made in identifying ‘risk’ variants for NTDs in a number of genes and some of these studies have identified risk factors that may show differing genetic predisposition among ethnic groups [2,40]. Potential variation in genetic predisposition among ethnic groups is suggested by differences in the NTD prevalence between ethnic groups, which in some cases persists after migration to other geographical locations. In the current study, we focused on a cohort from Malaysia. The Malaysian population is multi-ethnic, and this study included individuals of Malay, Chinese and Indian origin.

In particular, the Malaysian population seems biased towards the closed type of Neural Tube Defect seen in preliminary datasets ([41], unpublished data from the Invertebrate and Vertebrate Neurobiology Laboratory, Universiti Malaya). In the preliminary datasets, based on MRI and surgical reports [4], lipomyelomeningocele, lipomeningocele, and meningocele were classified as spina bifida occulta (closed type NTD) whilst myelomeningocele and meningocele were classified as spina bifida aperta (open type NTD). The potential relationship of open and closed spina bifida is further supported in genetic models of mouse knockouts which exhibit both open and closed neural tube defects in the same family of molecules [15,16,22]. Furthermore, there exists a number of mouse models which display spinal dysraphism, whose etiology remains unresolved as the phenotype was not ascertained during primary neurulation [42,43,44]. Therefore, in this study, both types of human spina bifida—4 occulta and 3 aperta—were recruited. The objective for this study was to screen pathogenic variants in *Eph* and *ephrin* genes through whole exome sequencing in a spina bifida cohort.

## 2. Materials and Methods

### 2.1. Proband Selection for Whole Exome Sequencing and Candidate Gene Validation

Seven unrelated individuals with sporadic spina bifida (5 Malays, 1 Chinese, and 1 Indian) and their healthy family members, where available, were recruited for this study. The 7 patients presented with spina bifida with neurological deficits, encompassing 4 spina bifida occulta and 3 spina bifida aperta. In total, 31 individuals participated in this study. The 7 patients (5 Malays, 1 Chinese and 1 Indian) with sporadic spina bifida were grouped into distinct families, comprising 5 complete trios (mother–father–proband), 1 quartet family (mother–father–twin sibling–proband), and 1 single-parent family (mother–proband) (Appendix A). The remaining 10 subjects (6 Malays, 2 Chinese, and 2 Indians) were healthy individuals unrelated to the probands and were included as comparable controls.

There were no descriptions of NTD cases in other family members and none of the patients were the result of consanguineous parents. An assumption was made based on the possibility that NTD phenotypes may be due to mutations in shared genes, thus NTD datasets were grouped based on type of spina bifida and similarity of neurological deficits (Appendix A).

Informed consent was obtained from all study participants. Collection of samples was performed in accordance with the ethical approval given by the Medical Ethics Committee (MEC) at the University of Malaya Medical Centre (UMMC) (MEC reference number: 914.5).

### 2.2. Whole Exome Sequencing

DNA was isolated from peripheral blood, saliva samples (Oragene^®^-DNA OG-500 manufactured by DNA Genotek^®^, Ottawa, ON, Canada) or buccal mouthwash of spina bifida individuals. Samples were prepared in Illumina TruSeq kit and Agilent SureSelect Target Enrichment kit for whole exome sequencing (WES). The samples were sequenced using the Illumina HiSeq 2000 and HiSeq 4000 platforms. The platforms were able to generate an average read length of 100 bp and had a median depth of 50x coverage per sample. Data alignment and variant analysis were performed [45,46]. The datasets generated and/or analyzed during the current study are available from the corresponding author on reasonable request.

### 2.3. Exome Datasets Analysis

The variants from the 7 probands of spina bifida were filtered for non-synonymous (missense), frameshift variants, and non-frameshift variants in the exonic region. The variants within the splicing region located adjacent to exons were also included. Variants with minor allele frequency (MAF < 0.01) based on 1000 Genome Phase 1 (annotated based on database build dbSNP135 with November 2010 and October 2011 allele frequency data) were first short-listed. The variant location and nucleotide changes were compared in probands and parents using the platform Galaxy biomedical data tool. Further, the variants were filtered based on a mapping quality of above 50 and a read depth higher than 15 to ensure that the variants were mapped to the reference genome with a high degree of confidence and reduced error probabilities [47,48]. The homozygous variants with alternate allele to total read depth ratio (AD/TD) above 85% and heterozygous variants range between 30% and 70% will be included to get the most accurate candidate variants [49].

Subsequently, the MAF of candidate variants were checked against 1000 Genome Phase 3 (annotated based on dbSNP 142 April 2020), gnomAD version 2.1.1 [50], ExAC, ESP and TOPMED. Candidate variants that are rare (MAF < 0.01), low frequency (MAF 0.01–0.05), and not reported in the 5 global databases were short-listed [51,52]. The MAF of variants in the specific population database, which were the Asian sub-database (gnomAD and ExAC), the East Asian (EAS) sub-database (1000 Genome Phase 3 and gnomAD), the South Asian (SAS) sub-database (1000 Genome Phase 3) and the Singaporean (Singapore Genome Variation Project (SGVP)) variant calling files (VCF) (genotype data build on human genome 18 project assembly, Hg18, March 2006) [53] were included for the inclusion of ancestry-matched controls [54].

Whole exome sequencing analysis was performed to identify variants in 22 known EphAs, EphBs, ephrinAs and ephrinBs genes. All candidate variants in 22 known EphA (*EPHA1–EPHA8*, and *EPHA10)*, *EphB (EPHB1–EPHB4,* and *EPHB6)*, *ephrinA (EFNA1–EFNA5)* and *ephrinB (**EFNB1–EFNB3)* genes were subjected to bioinformatics analysis to determine possible pathogenicity. The loss-of-function (LOF) effect was checked against *in silico* protein function and conservation tools: Polyphen-2 (Polymorphism Phenotype-2) [55], SIFT (Sorting Tolerance from Intolerance) [56], PROVEAN (Protein Variant Effect Analyser) [57], CADD [58], FATHMM-MKL [59], MutationTaster [60], GERP [61], and PhyloP [62,63]. The position of the variants and the predicted structure of the protein were analyzed using SMART (Simple Modular Architecture Research Tool).

### 2.4. PCR and Sanger Sequencing for Validation

PCR amplification and Sanger sequencing were used to verify candidate variants from the WES data and to analyze the variants in the family members (where available) and 10 controls (without NTD). A total of 3 sets of primers were designed (Appendix A).

## 3. Results

### 3.1. Whole Exome Sequencing Analysis of Eph and Ephrin Identifies 3 Variants

The possible contributions of the *Eph-ephrins* were investigated. There were no shared exonic variants in any known mammalian *Eph* and *ephrin* genes (*EPHA1–EPHA8*, *EPHA10*, *EPHB1–EPHB4*, *EPHB6*, *EFNA1–EFNA5*, and *EFNB1–EFNB3*) that fulfilled the MAF < 0.01 criteria in all seven spina bifida probands. Each exome dataset was screened individually in all seven probands to look for potential variants. We compared whole exome analysis on five complete trios (SB1A, SB2A, SB5A, SB7A, and SB13A), one quartet-families (SB27A) and one single-parent family (SB3A) and evaluated all possible means of transmission for the *Eph* and *ephrin* variant: de novo variants and homozygous and/or compound heterozygous variants for a recessive transmission. From 13 *Eph* and *ephrin* variants, there were no variants that represented de novo variants or emulated recessive transmission and appear to segregate exclusively with the phenotype (Appendix A).

Nevertheless, from the thirteen candidate variants, three variants in *EPHA2 (*rs147977279)*, EPHB6* (rs780569137) and *EFNB1* (rs772228172) were unreported or MAF < 0.01 and were predicted as pathogenic and conserved in at least seven *in silico* prediction tools. The *EPHA2* variant (rs147977279) was found in a heterozygous state in proband SB2A and the father of SB2A. The *EPHB6* variant (rs780569137) was found to be heterozygous in the exome of proband SB5A and the mother of SB5A. The *EFNB1* (rs772228172) was found to be heterozygous in the exome of proband SB1A and was genotyped as hemizygous in the father of SB1A (Table 1). Each of the candidate variants found in one of SB2A (rs147977279 in *EPHA2*), SB5A (rs780569137 in *EPHB6*) or SB1A (rs772228172 in *EFNB1*) were detected in single families but not in other spina bifida families or the control group.

The rs147977279 variant in the EphA-type receptor, *EPHA2*, is a G to C base change at 16,477,423 on chromosome 1, in an individual with spina bifida occulta, SB2A (Table 2). This alters the coding sequence, resulting in a leucine to valine substitution in the ligand binding domain of EPHA2 (Figure 1). The rs780569137 variant in EphB-type receptor *EPHB6* is an A to G base change at 142,562,247 on chromosome 7, in an individual with spina bifida aperta, SB5A (Table 2). This alters the coding sequence resulting in a tyrosine to cysteine substitution in the ligand binding domain of EPHB6 (Figure 1). The rs772228172 variant in ephrinB-type transmembrane ligand, *EFNB1* is a C to T base change at 68,049,626 on chromosome X, in an individual with spina bifida occulta, SB1A (Table 2). This alters the coding sequence, resulting in an arginine to tryptophan substitution in the signal peptide domain of EFNB1 (Figure 1).

### 3.2. Unreported and Rare Eph and Ephrin Variants

Among the three variants, MAF of rs780569137 (*EPHB6*) were not listed in the five global population databases with total number of alleles between 5008 to 282,912 alleles, suggesting that the variant is novel (Table 2). The rs147977279 (*EPHA2*) was considered rare (MAF < 0.01) in the five global databases of total alleles between 5008 and 251,420 alleles. Similarly, the MAF of rs772228172 (*EFNB1)* was considered very rare (MAF < 0.01) in gnomAD (MAF = 0.000014; number of alleles (2n) = 4 alleles), and TOPMED (MAF = 0.000008; 2n = 1 allele) in a total of 124,568 to 282,912 alleles. The MAF of rs772228172 (*EFNB1)* was not reported in the ExAC or ESP global population databases which include a total of 13,006 and 121,250 alleles, suggesting that the variant is novel in the population covered by the ExAC and ESP databases (Table 3).

From the databases surveyed, rs147977279 (*EPHA2*) was found at allele frequency MAF < 0.01 (rare) in the Asian population in gnomAD exome samples (total number of alleles = 49,008 alleles) or ExAC (total number of alleles = 25,142 alleles). Similarly, rs147977279 (*EPHA2*) was not reported in the East Asian sub-population (EAS) in 1000 Genome Phase 3 (total number of alleles = 1008), or gnomAD (total number of alleles = 1556 alleles). However, rs147977279 (*EPHA2*) was found at a low frequency of 0.011 among a total of 978 alleles in the South Asian sub-population (SAS) (Table 3). Likewise, rs772228172 (*EFNB1)* was found at allele frequency MAF < 0.01 in the Asian population in gnomAD exome samples (total number of alleles = 26,746 alleles). Interestingly, in the SGVP database, whose population group (98 Malays, 99 Chinese, and 95 Indians) is a closer representation of the ethnic origins of Malaysians, none of the three reported SNPs were listed in the genotype data built on the human genome 18 project assembly (Hg18; March 2006), suggesting the variants were principally absent in 292 Singapore individuals (Table 3).

### 3.3. In Silico Prediction of the Effect of the Variants on the Protein

The deleterious impact scores relating to the protein function of each variant were predicted as probably damaging or possibly damaging in Polyphen2 HumDiv with scores between 0.709 and 1.00, where 1.00 is predicted to be the most damaging. In SIFT, the variants were predicted as damaging with scores between 0.000 and 0.001 with 0.05 as the cut-off value for variants with a deleterious impact. In CADD prediction analysis, the variants were ranked among the 1% most deleterious in the genome with scores above 20 as a cut-off value (Table 4). Similarly, in Polyphen2 HumVar and FATHMM-MKL, rs147977279 (*EPHA2*) and rs780569137 (*EPHB6*) were predicted to be damaging with high deleterious impact scores (between 0.94 and 0.997) compared to rs772228172 (*EFNB1)*’s scores (between 0.61563 and 0.709), whereby 1.00 is predicted to be the most damaging or pathogenic.

In MutationTaster, rs147977279 (*EPHA2*) was predicted as probably deleterious (disease causing) whereas rs780569137 (*EPHB6*) and rs772228172 (*EFNB1)* were predicted to be probably harmless (polymorphism). In Provean, all variants were predicted as neutral with scores ranging between −0.51 and −2.27 with a cut-off value of −2.5. In sequence evolutionary conservation analysis, the wild-type nucleotide at c.G121 for rs147977279 (*EPHA2*), c.A689 for rs780569137 (*EPHB6*) and c.C7 for rs772228172 (*EFNB1)* are highly constrained with GERP scores of 4.17 to 5.21 (cut-off value is −12.36 to 6.16), and PhyloP scores of 1.538 to 3.28894 (cut-off value is −14 to 6) (Table 4).

### 3.4. Spina Bifida-Related Genes in Probands with Eph and Ephrin Variants

This study extends the investigation into the possible involvement of spina bifida-related genes (Appendix A, [4,10]) amongst the datasets with *Eph* and *ephrin* variants which we validated in SB1A, SB1C, SB2A, SB2C, SB5A and SB5B. Thirty selected variants in reported spina bifida-related genes were short-listed in Table 5. The lists were filtered based on variants that segregate with spina bifida in probands (Table 5, Row A) and variants that had different genotype annotation in probands and parent of probands with the *Eph* and *eph* variants (Table 5, Row B). In the latter list, the pattern of the *Eph-ephrin* variants from this study and the spina bifida-related gene were compared in probands (SB1A, SB2A, and SB5A) and parents of probands with the *Eph-ephrin* variants (SB1C, SB2C, and SB5B).

In the exome dataset of proband SB2A with the *EPHA2* variant (rs147977279), four variants in *ALDH1L1* (rs2886059C>A)*, CUBN* (rs62619939C>G)*, GRHL3* (rs2486668C>G) and *PARD3* (rs118153230C>T) were de novo variants with genotypes in both father (SB2C) and mother (SB2B) of proband SB2A and were wildtype (Table 5). Specifically, the allele frequency of the *PARD3* variant (rs118153230C>T) was considered rare (MAF < 0.01) and two other variants in *ALDH1L1* (rs2886059C>A) and *CUBN* (rs62619939C>G) were predicted with a damaging effect on the protein function (based on PolyPhen2 HumDiv, PolyPhen 2 HumVar, SIFT, Provean, and CADD; Appendix A). On the other hand, three variants in *BHMT* (rs3733890G>A), *MTHFD1* (rs2236225G>A) and *TNIP1* (rs2233311C>A) that did not comply with the de novo pattern were predicted to have a damaging effect on the protein function.

Likewise, in the exome dataset of proband SB5A with *EPHB6* variant (rs780569137), five variants in spina bifida-related genes were likely pathogenic based on MAF and *in silico* predictions (Table 5, Appendix A). One of the five variants was found in *MTRR (*rs1801394A>G) and was annotated as homozygous in proband SB5A and by contrast as heterozygous in SB5B (mother of proband SB5A with *EPHB6* variant). The *MTRR* variant (rs1801394A>G) was predicted to have a damaging effect on the protein function. Another four variants were found in *SCRIB* (rs781978489G>A), *MTHFR (*rs200947520G>T, and rs1801133G>A) and *XPD (*rs1799793C>T) and were annotated as heterozygous in proband SB5A, and as wildtype in SB5B. Notably, the *SCRIB* variant (rs781978489G>A) was considered a rare variant and was predicted to have a damaging effect on the protein function (Appendix A). Other than the *SCRIB* (rs781978489G>A) and *MTRR (*rs1801394A>G) variants, the allele frequency of *MTHFR* (rs200947520G>T) was considered rare and two other variants in *MTHFR* (rs1801133G>A) and *XPD* (rs1799793C>T) were predicted to have a damaging effect on the protein function. Remarkably, de novo variants were not found in the exome dataset of proband SB5A (Table 5).

As in the exome dataset of proband SB1A with the *EFNB1* variant (rs772228172), four variants were annotated as homozygous (*CUBN, COMT, PTCH1* and *TRDMT1)* and eight variants as heterozygous (*CUBN,*
*MTRR,* and *VANGL1*) in the reported spina bifida-related genes (Table 5). However, no comparison could be made as the exome dataset was not available for the father of proband SB1A (SB1C) with the *EFNB1* variant (rs772228172). Among the twelve variants listed as potential de novo variants (if the variants for SB1C were validated), one variant in *VANGL1 (Chr01:116206438C>A)* was unreported (novel) as well as predicted to have a damaging effect on protein function (Table 5, Appendix A). Other than the variant in *VANGL1*, two other variants in *CUBN* (rs143400113C>T, and rs369981313G>T) were considered rare (MAF < 0.01) and five other variants in *PTCH1* (rs357564G>A), *CUBN* (rs1801224G>T, and rs2271460A>C), and *MTRR* (rs1801394, and rs2287780C>T) were predicted to have a damaging effect on protein function.

## 4. Discussion

Based on whole exome sequencing and subsequent validation by PCR, nearly half of the Malaysian spina bifida families (three out of seven families) analyzed in this study were found to carry variants in *Eph* or *ephrin* encoding genes. In each of the families, other than the proband, at least one unaffected member was detected as a heterozygous or hemizygous carrier of the variant allele, suggesting an uncaptured additive gene–gene or gene–environmental interaction that potentially disrupted neural tube closure in the probands.

Based on the interrogation of the 1000 Genome Phase 3, gnomAD, ExAC, ESP and TOPMED databases, these variants are novel or rare with an unreported allele (novel) or MAF of 0.0024 and below (Table 3), based on the accepted convention that any variants recorded in less than 1% or 0.01 of the general population are considered to be rare [52]. Furthermore, *EPHB6* (rs780569137) and *EFNB1* (rs772228172) variants were not reported, not listed (based on variant calling files, vcf) or rare (MAF < 0.01) in the population specific databases (East Asian, South Asian and Singaporean). The Singaporean database (SGVP) is especially relevant, as the population group genotyped consisted of 292 Singaporean Malay, Chinese and Indian individuals who are a closer representation in genetic ancestries to the Malaysian individuals in our study. The *EPHA2* variant (rs147977279) is reported to have low frequency (0.01 < MAF < 0.1) in the South Asian population (1000 Genome Phase 3 database) but the variant was not listed in the Singaporean database (SGVP). Thus, the reported rs147977279 (*EPHA2*) allele number (2n = 978 alleles) in the South Asian subpopulation (SAS) was perhaps unlikely to have been identified in Singaporean or Malaysian individuals (Table 2).

By considering the potential contributing functions of Ephs and ephrins in neural tube closure [15,16,22], and the in silico prediction of the effect of the variants on the protein (in seven to eight out of the nine prediction tools), we suggest that these variants may play a causative role in the phenotypes. Moreover, the variants scored high in terms of the deleterious impact of its protein function and are remarkably close to the cut-off values (scores 0.94351 to 1.00 for cut-off value 1.00; scores 0.000 to 0.001 for cut-off value 0.05) indicating the high probability for pathogenicity by amino acid substitutions (Table 4). However, the prediction that the *EPHB6* and *EFNB1* variants are probably harmless in FATHMM-MKL might be due to differences in the categorization and the algorithm’s predefined threshold scores compared to other prediction tools [64]. In PROVEAN, all three variants were predicted to have a neutral effect on the protein function, which might be due to the implicit assumption in the PROVEAN method that large changes to proteins are deleterious (the more dissimilar to the query sequence the worse the score) [57,65]. In this way the PROVEAN scoring algorithm is more generic in evaluating the amino acid sequence (multiple amino acids) compared to others, such as PolyPhen2 and SIFT, which are more specific at the position of interest (single amino acid) [55,56,57,63]. Ultimately, functional studies will be required to confirm the variant pathogenicity on the neural tube development and spina bifida susceptibility.

Eph-ephrin proteins such as EphA7 [15] and ephrinB1 [17], and *Eph-ephrin* mRNAs such as *ephrinA5* [15], *ephrinA1, ephrinA3, EphA1, EphA2, EphA4, EphA5* [16] and *EphA7* (Wang et al., 2016) have been reported to be expressed in the neural tubes of mouse and *Xenopus* embryos. Specifically, *EphA2* was found to be expressed in surface ectoderm and lamellipodium-like protrusions at the neural fold tips of E9.5 mouse embryos [16]. EphrinB1 was detected in the neuroepithelial cells of E10.5 mouse embryos [17]. However, an association of EphB6 function with neural tube development has not been reported in human or animal model embryos, to date.

Based on the exome datasets, variants in a set of genes relating to environmental factors, and planar cell polarity genes were also found in the three probands with *Eph* and *ephrin* variants (Table 5). It would be interesting to further investigate potential gene–gene interactions of variants in *Eph* and *ephrin* genes and spina bifida-related genes. For instance, there were three de novo variants with allele frequency that were considered rare (*PARD3 rs118153230C>T*) or were predicted to have a damaging effect on protein function (*ALDH1L1 rs2886059C>A*, and *CUBN rs62619939C>G*) found in proband SB2A with the *EPHA2* variant (rs147977279) (Table 5 and Appendix A). On the other hand, it is interesting that in the proband with a severe type of spina bifida, such as spina bifida aperta (SB5A), de novo variants were not found in comparison to proband SB2A that were diagnosed as spina bifida occulta. However, larger cohorts of individuals with spina bifida occulta and aperta will be required for a conclusive comparison.

Although the exome dataset of the father of proband SB1A (SB1C) was not available to screen for a de novo or recessive inheritance pattern, it would be interesting to further test if the novel variant in *VANGL1* (Chr01:116206438C>A; planar cell polarity gene) that was found in proband SB1A and predicted to have a damaging effect on the protein, could play an additive or combinatorial role with the *EFNB1* variant (rs772228172) in spina bifida susceptibility. Another variant in *VANGL1* (rs4839469) was also detected in proband SB2A and proband SB5A. This variant had a different genotype in the probands with the *Eph* variants (*EPHA2,* and *EPHB6)* compared to their parents. Furthermore, another variant in *SCRIB* (rs781978489G>A), which is also a planar cell polarity gene, was also found in proband SB5A. The *SCRIB* (rs781978489G>A) variant had a rare allele frequency and predicted a damaging effect on the protein as well.

Among the three probands with *Ephs* and *ephrins* variants, probands SB1A and SB5A both have variants in *Ephs* and *ephrins,* and *MTRR* (rs1801394) (Table 5). Moreover, in exome datasets of proband SB2A and the father of proband SB2A (SB2C), who are both heterozygous for *EPHA2* variant (rs147977279), two different variants in *MTRR* (rs162036, and rs10380) were also annotated as heterozygous in SB2A but not in SB2C (wildtype). Our previous study on the same cohort revealed that 57% of patients and 83% of parents carried the rs1801394 variant in their *MTRR* gene, based on either homozygous (G/G) or heterozygous (A/G) genotypes [66]. In that study, we concluded that the *MTRR* rs1801394 variant may be an NTD risk factor in the Malaysian population based on the prevalence of this variant in other studies and that its association with NTDs differed across populations worldwide [66]. Hereto, the polymorphisms in *Eph* and *ephrin* genes, single one-carbon metabolism genes or planar cell polarity genes could play a limited role in overall NTD risk determination [10,67,68,69].

## 5. Conclusions

In summary, we report the existence of variants in *Eph* and *ephrin* genes in three out of seven NTD families in Malaysia. Even though the sample size in this study is small, it provides initial evidence of the need to screen for these predicted pathogenic variants in a larger Malaysian cohort. In each of the families, other than the proband, at least one unaffected member was detected as a heterozygous or hemizygous carrier of the variant allele, suggesting *Eph* and *ephrin* variants as a potential screening tool for NTD development among Malaysians. Further validation experiments at the transcript level and a gene perturbation study would help to assess the gene–gene interactions and gene function to support the pathophysiology of NTD etiology amongst *Eph* and *ephrin* candidate genes.

## Figures and Tables

**Figure 1 genes-13-00952-f001:**
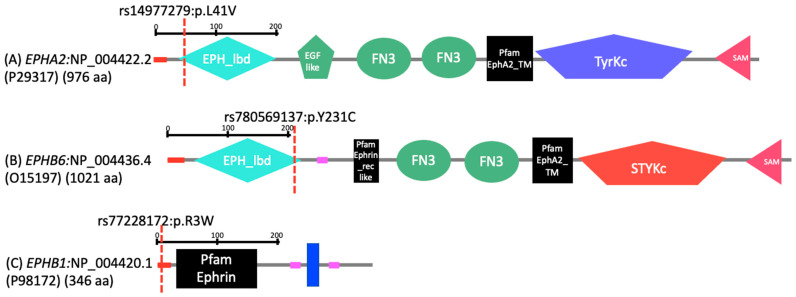
The EPHA2, EPHB6 and EFNB1 variants in their protein form. The position of the non-synonymous variants (EPHA2:rs147977279, EPHB6:rs780569137 and EFNB1:rs772228172) were represented by red dashed bar (L = Leucine; V = Valine; Y = Tyrosine; C = Cysteine; R = Arginine; W = Tryptophan; EPH_lbd = Eph ligand binding domain; EGF like = epidermal growth factor-like; FN3 = fibronectin-3; Pfam EphA2_TM = protein family of EphA2 transmembrane; TyrKc = tyrosine kinase catalytic domain; SAM = sterile alpha domain; Pfam Ephrin_rec like = protein domain tyrosine-protein kinase ephrin type A/B receptor like; STYKc = Protein Ser/Thr/Tyr kinase or phosphotransferases; Pfam ephrin = protein family ephrin; red box = signal peptide; blue box = transmembrane region; pink box = low complexity region).

**Table 1 genes-13-00952-t001:** Genotypes of spina bifida probands from exome datasets.

Probands	Genotypes						
	*EPHA2* (rs147977279)		*EPHB6* (rs780569137)		*EFNB1* (rs772228172)		
	G/G	G/C (Heterozygous)	A/A	A/G (Heterozygous)	C/C	C/T (Heterozygous)	T (Hemizygous)
Proband	-	1 (SB2A)	-	1 (SB5A)	-	1 (SB1A)	-
Parent of probands	1 (Mother of SB2A)	1 (Father of SB2A)	1 (Father of SB5A)	1 (Mother of SB5A)	1 (Mother of SB1A)	-	1 (Father of SB1A)
Other NTD Probands	6	-	6	-	6	-	-
Parents and unaffected twin-sibling of other NTD probands	12	-	12	-	12	-	-
Controls	10	-	10	-	10	-	-
Total	29	2	29	2	29	1	1

**Table 2 genes-13-00952-t002:** Position and changes induced by the 3 gene variants.

Gene	Location	Transcript	cDNA Change	Amino Acid Change	Nucleotide Change
*EPHA2* rs147977279	Chr1: 16477423	NM_004431 (exon 2)	c.G121C	p.L41V	G to C
*EPHB6* rs780569137	Chr7: 142562247	NM_004445 (exon 7)	c.A689G	p.Y230C	A to G
*EFNB1* rs772228172	ChrX: 68049626	NM_004429 (exon 1)	c.C7T	p.R3W	C to T

**Table 3 genes-13-00952-t003:** Minor Allele Frequency (MAF) databases information specific for *EPHA2* (rs147977279), *EPHB6* (rs780569137), and *EFNB1* (rs772228172).

MAF Databases	*EPHA2* (rs147977279)	*EPHB6* (rs780569137)	*EFNB1* (rs772228172)
1000 Genome Phase 3 (2N = 5008 alleles)	Global: 0.0022 (2n = 11) EAS = Not reported (total number of alleles = 1008) SAS = 0.011 (2n = 978)	Not listed	Not listed
GnomAD v2.1.1 (2N = 282,912 alleles)	GnomAD exomes (Global) = 0.002013 (2n = 506/251,420) GnomAD exomes (Asian) = 0.00853 (total number of alleles = 49,008) GnomAD (Global) = 0.000019 (2n = 6/31,384) GnomAD (EAS) = Not reported (2n = 0/1556)	Not listed	GnomAD exomes (Global) = 0.000014 (2n = 2/142,298) GnomAD exomes (Asian) = 0.00004 (total number of alleles = 26,746)
ExAC (2N = 121,250 alleles)	Global = 0.0024 (2n = 291) Asian = 0.00927 (total number of alleles = 25,142)	Not listed	Not reported
ESP (2N = 13,006 alleles)	Global = 0.0000308 (2n = 4/13,006)	Not listed	Not reported
TOPMED (2N = 124,568 alleles)	Global = 0.000334 (2n = 42/125,568)	Not listed	Global = 0.000008 (2n = 1/125,568)
Singapore Genome Variation Project (SGVP)	Not listed	Not listed	Not listed

2n is allele count of variants in 1 individual (n); EAS = East Asian population; SAS = South Asian population; Not listed = the reported SNP (rs) were not called in variant call database (VCF); Not reported = The reported SNP (rs) is not found (2n = 0) in the database.

**Table 4 genes-13-00952-t004:** Predictions of pathogenicity potential and sequence conservation analysis of *Eph* and *ephrin* candidate variants.

Function	Tool	Score Cut-Off/Range	*EPHA2* (rs147977279)		*EPHB6* (rs780569137)		*EFNB1* (rs772228172)	
			Score	Prediction	Score	Prediction	Score	Prediction
Pathogenicity	Polyphen2 HumDiv	0–1	0.998	Probably Damaging	1.00	Probably Damaging	0.999	Probably Damaging
	Polyphen2 HumVar	0–1	0.997	Probably Damaging	0.962	Probably Damaging	0.709	Possibly Damaging
	SIFT	Cut-off = 0.05	0.001	Damaging	0.000	Damaging	0.001	Damaging
	Provean	Cut-off = −2.5	−2.27	Neutral	−1.92	Neutral	−0.51	Neutral
	CADD	0–10 = Bottom 90% 10–20 = Top 10% >20 = Top 1%	25.2	Top 1% most deleterious in the genome	24.0	Top 1% most deleterious in the genome	32	Top 1% most deleterious in the genome
	FATHMM-MKL	0–1	0.94351	Pathogenic	0.94876	Pathogenic	0.61563	Pathogenic
	MutationTaster	-	Disease Causing	Probably deleterious	Polymorphism	Probably harmless	Polymorphism	Probably harmless
Sequence conservation	GERP	−12.36 to +6.16	4.78	Evolutionary constrained	5.21	Evolutionary constrained	4.17	Evolutionary constrained
	PhyloP	−14 to +6	3.25157	Conserved	3.28894	Conserved	1.538	Conserved

**Table 5 genes-13-00952-t005:** The predicted impact of the spina bifida-related variants on the potential interaction with the *Eph* and *ephrin* candidates in probands.

	SB1A	SB1C (Father of Proband SB1A)	SB1B (Mother of Proband SB1A)	SB2A	SB2C (Father of Proband SB2A)	SB2B (Mother of Proband SB2A)	SB5A	SB5B (Mother of Proband SB5A)	SB5C (Father of Proband SB5A)
*Ephs* and *ephrins*	*EFNB1* (het)	*EFNB1* (hemi)	*EFNB1 (wt)*	*EPHA2* (het)	*EPHA2* (het)	*EPHA2 (wt)*	*EPHB6* (het)	*EPHB6* (het)	*EPHB6 (wt)*
(A) Reported spina bifida-related genes with variants segregated with spina bifida	****^/^****PTCH1* (hom)** * *CUBN.1* (het) ** *CUBN.2* (het) *CUBN.3* (het) ** *CUBN*.4 (het) * *CUBN*.5 (het) ** *MTRR*.1 (het) ** *MTRR.4* (het) *^/^** *VANGL1.1* (het) ***** COMT (hom)** ****** CUBN.6*** **(hom)** ****** TRDMT1*** **(hom)**	Exome dataset not available	***PTCH1 (*****wt)** ***CUBN.1*****(wt)** ***CUBN.2*****(wt)** ***CUBN*****.3 (wt)** ***CUBN*****.4 (wt)** ***CUBN*****.5 (wt)** ***MTRR*****.1 (wt)** ***MTRR.4*****(wt)** ***VANGL1.1 (wt)*** COMT (het) *CUBN.6* (het) *TRDMT1* (het)	** *ALDH1L1* (het) ** *CUBN.*7 (het) *GRHL3* (het) * *PARD3* (het)	** *ALDH1L1* ** **(wt)** ** *CUBN.* ** **7 (wt)** ** *GRHL3* ** **(wt)** ** *PARD3* ** **(wt)**	** *ALDH1L1* ** **(wt)** ** *CUBN.* ** **7 (wt)** ** *GRHL3* ** **(wt)** ** *PARD3* ** **(wt)**	No variants found	No variants found	No variants found
# (B) Reported spina bifida-related genes in probands and parent of probands with *Ephs* and *ephrins* variant	Not relevant due to exome dataset SB1C was not available	Exome dataset not available	Not relevant due to exome dataset SB1C was not available	** *BHMT (*het) ** *MTHFD1* (het) *MTRR*.2 (het) *MTRR*.3 (het) ** *TNIP1* (het) *VANGL1.2* (het) ****** CUBN.6 (*****hom)** ****** SOD2*** **(hom)**	***BHMT*****(wt)** ***MTHFD1*****(wt)** ***MTRR*****.2 (wt)** ***MTRR*****.3 (wt)** ***TNIP1*****(wt) *VANGL1.2* (wt)** *CUBN.6 (*het) *SOD2* (het)	** *BHMT* (het) ** *MTHFD1* (het) *MTRR*.2 (het) *MTRR*.3 (het) ** *TNIP1* (het) ****** VANGL1.2*****(hom)** ****** CUBN.6*** **(hom)** ****** SOD2*** **(hom)**	*APEX1* (het) * *MTHFR.2* (het) ** *MTHFR.1* (het) *PCMT1* (het) *^/^** *SCRIB* (het) *VANGL1.2* (het) ** *XPD* (het) *ALDH1A2* (het) ****^/^***** ***MTRR*****.1 (hom)**	***APEX1*****(wt)** ***MTHFR.2*****(wt)** ***MTHFR.1*****(wt)** ***PCMT1*****(wt)** ***SCRIB*****(wt)** ***VANGL1.2*****(wt)** ***XPD*****(wt)** ***ALDH1A*****(wt)** *MTRR*.1 (het)	*APEX1* (het) * *MTHFR.2 (*het) ** *MTHFR.1 (*het) *PCMT1 (*het) *^/^** *SCRIB (*het) *VANGL1.2 (*het)** *XPD* (het) ****** ALDH1A2 (*****hom****)** ****^/^***** ***MTRR*****.1 (hom)**

The rs number of the variants were *EFNB1* = rs772228172C>T, *EPHA2* = rs147977279G>C, and *EPHB6* = rs780569137A>G, whereas the rs numbers for variants found in the reported spina bifida-related genes were represented in the following alphabetical order as: *ALDH1A2* = rs4646626C>T, *ALDH1L1 =* rs2886059C>A, *APEX1 =* rs1130409T>G, *BHMT =* rs3733890G>A, *COMT*= rs4680G>A, CUBN.1 = rs143400113C>T; CUBN.2 = rs1801224G>T, *CUBN.*3 = rs3740168G>C, *CUBN.*4 = rs2271460A>C, *CUBN.*5 = rs369981313G>T, *CUBN.6 =* rs1801231G>A, *CUBN*.7: rs62619939C>G, *GRHL3* = rs2486668C>G, *MTHFD1* = rs2236225G>A, *MTHFR.1* = rs1801133G>A, *MTHFR.2 =* rs200947520G>T, *MTRR*.1 = rs1801394A>G, *MTRR*.2 = rs162036A>G, *MTRR*.3 = rs10380C>T, *MTRR.4* = rs2287780C>T, *PARD3* = rs118153230C>T, *PCMT1 =* rs4816G>A, *PTCH1* = rs357564G>A, *SCRIB =* rs781978489G>A, SOD2 = rs4880A>G, *TNIP1* = rs2233311C>A, *TRDMT1* = rs11254413G>A; *VANGL1.2* = rs4839469G>A, *VANGL1*.1 = Chr01:116206438C>A, and *XPD =* rs1799793C>T; # Although, the variants were not segregated with NTD, however, the genotype of the parents with *Ephs* and *ephrins* was wildtype and heterozygous compared to the proband; * The predicted impact of the variants on the likely interaction with the *Ephs* and *ephrins* candidates from this study was annotated for MAF < 0.01 (gnomAD public database); ** Predicted as deleterious in one of the five *in silico* predictions. An extended list of MAF and five protein function predictions are available in Appendix A). wt/bold font = wildtype/reference genotype; *** hom/bold font = homozygous genotype; het = heterozygous genotype).

## Data Availability

The data presented in this study are available on request from the corresponding author. The data are not publicly available due to concerns regarding participant/patient anonymity.

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
