# Peer review of "Eph and Ephrin Variants in Malaysian Neural Tube Defect Families"

_genes, 2022, doi:10.3390/genes13060952_

Round 1

Reviewer 1 Report

These authors performed whole exome sequencing in 7 spina bifida cases in Malaysia and found variants in ephrin-EPH genes as potential candidate risk factors for spina bifida. One good thing about this study, is that these variants were not de novo variants, suggesting that rare, potentially damaging variants in ephrins-EPHs could be a valuable genetic screening tool for NTD risk prior to pregnancy. Its also good that they looked into potential gene-gene interactions with other NTD candidate pathways, as this is the type of information that would be most valuable in assessing risk.

In the abstract, sentence 2 - "Despite many gene candidates identified in certain populations; none have thus far been regarded as compelling enough as a candidate." This sentence is somewhat problematic. It's true that many NTD candidates have been identified in human genome/exome  studies, and that none of them can really be pointed to as singularly causative by conventional standards given that most of these variants are rare, heterozygotes, or even singletons occurring in the context of a rare disease, with a limited population for study. But they are compelling in the sense that they are in genes that, if knocked out in animal models, result in NTD phenotypes. Furthermore, in some studies follow up functional analyses of identified variants have demonstrated potential mechanisms of pathogenesis in the past , like failure to localize to the cell membrane in the case of LRP6 variants, or failure to localize to mitochondria in the case of FKBP8 variants. And often we generally find these loss-of-function variants in a heterozygous context, while homozygous loss-of-function is required for NTD phenotypes in animal models. But that only points to the importance of gene-gene-enviroment interactions in the etiology of these defects. The authors definitely acknowledge this point later in the paper, but this sentence in the abstract implies that their study has identified more compelling candidates than candidates identified in previous studies, when in fact this study is on par with other spina bifida genomic studies. So it'd be nice to see this sentence revised to more accurately introduce the abstract.

Overall, the manuscript is well written, but may need to be looked over for correction of english grammar, especially use of articles and pluralization.

line 57 regarding MTHFR C677T - May want to clarify that you are referring to homozygous individuals for this allele.

Introduction line 117-121. This sentence implies that their results were 'comparing' between occulta and aperta phenotypes, when really it was just that they included both aperta and occulta patients in their study. I agree that they should include both and find their argument for doing so in the Malaysian population compelling. It would be preferable if the "unpublished data" cited on line 121 could be scrutinized to validate their claim, but it is likely the case that a majority of spina bifida cases in all populations are less severe, occulta phenotypes as these often go undiagnosed, are asymptomatic, and are often not included in these studies. I do appreciate them delineating the heterogeneity in spina bifida presentation among their study participants in Table S2.

Please correct 'close' to 'closed' throughout referring to closed neural tube defect phenotypes (aka occulta).

It may be helpful if the authors could include a figure or supplementary figure summarizing their cohort as outlined in line 132-140. Labelled pedigrees for each trio, triad, etc would be helpful for the reader to visualize which proband was part of which trio, ethnic group, etc.

There appears to be a sentence from the submission system or guide for authors template accidentally pasted into the manuscript on lines 196-198.

Figure 1 probably needs to be cleaned up a little and provided at a higher resolution. The text in the Pfam domains are especially hard to read.

I would like to see more info on the de novo variants listed in table 5 that are postulated as being interacting risk factors. A supplementary table with polyphen/SIFT scores etc. Information that may allow others to develop hypotheses regarding mechanisms of the gene-gene interactions that could be tested in other studies.

Overall, it's worth publishing information on these variants. Being able to screen for variants in these genes or study gene-gene interacting mechanisms between ephrin-EPH genes and other NTD candidates may ultimately lead to some prevention of these birth defects in Malaysia and elsewhere as healthcare systems move toward precision prevention strategies.

Reviewer 2 Report

The authors reported the presence of Eph and ephrin gene variants prevalention in a small cohort of spina bifida patients in Malaysian families.

Quality of data is good, images are clear and representative, results are consistent with the conclusions. However, some aspects need to be elucidated:

  1. How do you explain neutral prediction in Provean in silico analysis?
  2. The limitations of the study need to be discussed.
  3. Is there any immunohistological evidence of Epf and ephrin appearance in neural tissue or other ectodermal derivates?
